# Association between Pb, Cd, and Hg Exposure and Liver Injury among Korean Adults

**DOI:** 10.3390/ijerph18136783

**Published:** 2021-06-24

**Authors:** Do-Won Kim, Jeongwon Ock, Kyong-Whan Moon, Choong-Hee Park

**Affiliations:** 1Environmental Health Research Division, National institute of Environmental Research, Ministry of Environment, Incheon 22689, Korea; ehdnjs8334@korea.kr (D.-W.K.); ockjeongwon@korea.kr (J.O.); 2BK21 FOUR R & E Center for Learning Health System, Department of Health and Environmental Science, Korea University, Anam-ro 145, Seongbuk-gu, Seoul 02841, Korea; kwmoon@korea.ac.kr

**Keywords:** heavy metals, liver function, KoNEHS, epidemiology

## Abstract

Background: Exposure to lead (Pb), cadmium (Cd), and mercury (Hg) has been reported to be associated with liver-related diseases. However, studies examining the association between heavy metal exposure and liver injury in a large population are scant and characterized by inconsistent results. This study aimed to evaluate the association between levels of heavy metal exposure and liver injury in the general population. Methods: Data for 2953 participants aged 19 years or more obtained from the Korean National Environmental Health Survey (KoNEHS) Cycle 3 (2015–2017) were used. The associations between levels of blood or urine heavy metals (Pb, Cd, and Hg) and liver function biomarkers [aspartate aminotransferase (AST), alanine aminotransferase (ALT), and gamma-glutamyl transferase (GGT)] were evaluated using multiple linear regression analysis. Results: Regarding the blood Pb (BPb), AST was higher in those of the 4th quartile, ALT was higher in those of the 2nd and 4th quartiles, and GGT was higher in those of the 3rd and 4th quartiles than in the 1st quartile. For urinary Cd (UCd), AST was higher in those of the 4th quartile; ALT was higher in those of the 2nd, 3rd, and 4th quartiles; and GGT was higher in the 4th quartile than in the 1st quartile. For the blood Hg (BHg), AST was higher in those of the 2nd and 4th quartile, ALT was higher in those of the 2nd, 3rd, and 4th quartiles; and GGT was higher in those of the 3rd and 4th quartiles than in the 1st quartile. There was no significant difference between urinary Hg (UHg) and liver function markers. Conclusion: Heavy metal exposure showed an association with liver injury among the general adult population in Korea. Further studies are required to clarify the relationship between heavy metals and liver injury.

## 1. Introduction

Heavy metals are emitted into the environment from both natural and industrial activities [1]. Lead (Pb) is released from exhaust gas, paints, and industrial wastes [1,2]; cadmium (Cd) is released from batteries, pigments, coatings, and plating substances [3]; and mercury (Hg) is released from thermometers, bulbs, dental fillers, pharmaceutical products, and pesticides [4]. In general, ingestion, inhalation, and dermal absorption are the main routes of heavy metal exposure to humans [5]. Heavy metals have the properties of accumulating in selected tissues of the human body and the potential to be toxic even at low levels of exposure [6]. Heavy metal exposure was associated with nonalcoholic fatty liver (NAFLD), nonalcoholic steatohepatitis (NASH), and hepatic fibrosis [7,8].

The liver metabolizes heavy metals and then excretes them into the intestines through bile [9,10]. About 5% of the substance is removed through the feces and 90–95% can be reabsorbed by the enterohepatic circulation [11,12]. As a result, liver cells are exposed to chemicals, which can lead to liver dysfunction, cell damage, and organ failure [13,14,15,16]. Considering that aspartate aminotransferase (AST), alanine aminotransferase (ALT), and gamma-glutamyl transferase (GGT) from damaged hepatocytes are released into the blood, these are useful markers for liver injury [17,18,19]. An increase in serum AST is generally observed in acute hepatocellular injury, and serum ALT is considered a specific indicator of hepatocyte injury because it has a low concentration in non-liver tissues [20,21]. Serum GGT is commonly used as a marker of oxidative stress and liver diseases [20,21,22,23]. Associations between exposure to heavy metals and liver injury have been examined in several studies, but the results are inconsistent. According to Obeng-Gyasi [24], serum AST, ALT, and GGT concentrations are increased by exposure to low levels of Pb in US adults, while in Can et al. [25] reported no significant difference in liver function biomarkers between 60 battery or muffler repair workers exposed to Pb and 24 controls. Further, Park et al. [14] reported an increase in serum ALT and GGT concentrations as Cd exposure was associated with increased ALT in 12,099 Korea adults, while an in vivo study on broilers that were exposed to Cd through ingestion showed no association with elevated serum AST, ALT, and GGT concentrations [26]. Cave et al. [7] reported a positive correlation between Hg exposure and serum ALT in 4582 adult participants in the US. But, Poursafa et al. [27] reported that AST and ALT concentrations did not increase with Hg exposure in 320 adolescents. However, heavy metal toxicity can be attributed to a variety of sources, including the dose, exposure route, and chemical species and sex, age, race, genetics, and nutritional factors related to the individual [28]. 

Therefore, it is necessary to determine the effect of heavy metals on liver injury in adults representing the general population. We aimed to investigate the association between blood or urine concentrations of Pb, Cd, and Hg with liver function biomarker [aspartate aminotransferase (AST), alanine aminotransferase (ALT), and gamma-glutamyl transferase (GGT)] concentrations using data from the Korean National Environmental Health Survey (KoNEHS) Cycle 3 (2015–2017). The present study incorporates a nationally representative general population of Korea with the largest population size reported to date. The results of this study can confirm the adverse liver function effects of heavy metals among the general adult population.

## 2. Materials and Methods

### 2.1. Study Population

KoNEHS is a nationwide cross-sectional study representing the Korean adult population designed to estimate exposure levels of environmental pollutants and the factors associated with these exposures. For this study, a two-stage proportional stratified sampling design was developed based on sex, age, and geographic characteristics. Additional information on the study design is available in [29]. During the KoNEHS Cycle 3, 15 households were selected for sampling using a systematic method from eligible households excluding foreign households in 233 sample enumeration districts, with household members aged ≥19 years selected as participants; all household members aged ≥19 years in the sample households were selected as subjects for the survey. During the survey period, about 3787 adult participants aged ≥19 years were recruited. Urine and blood samples were collected by medical technicians and nurses under the supervision of doctors, and demographic and socioeconomic data were collected through face-to-face interviews. We excluded participants who self-reported taking personal medications for chronic hepatitis, acute liver disease, fatty liver, and liver cancer (*n* = 38). We also excluded participants who were not measured for heavy metals or liver biomarkers (*n* = 47) and urinary creatinine (*n* = 3). Furthermore, we excluded participants who had an abnormal liver function range (*n* = 746). As a result, 2953 subjects were used for the analysis. This study was approved by the Institutional Review Board of the National Institute of Environmental Research [30].

### 2.2. Heavy Metals Measurement 

Blood samples were collected from the participants according to the vacutainer blood collection method, and spot urine samples were collected. The samples were kept frozen at −20 °C until analysis. Blood Pb (BPb) and urinary Cd (UCd) were analyzed using a graphite furnace-atomic absorption spectroscopy system (AAnalyst-800, Perkin Elmer, Norwalk, CT, USA). Blood Hg (BHg) and urinary Hg (UHg) were analyzed using a gold amalgamation direct mercury analyzer (DMA-80, Milestone, Sorisole, Italy). For external quality control, all of the analytical laboratories participated in the G-EQUAS and Special Health Quality Control programs. The results from all laboratories passed the G-EQUAS and Special Health Quality Control programs of Korea. To ensure the interval validity of clinical test substance analysis, a code of conduct regarding method detection limits (MDL), accuracy, and batch-specific precision was established in advance [31]. Accuracy is managed within ±2SD of the average of the QC sample results, and the precision is managed within 10% of the relative standard deviation of the QC sample analysis results. Heavy metals were all managed within the internal quality control standards. The limit of detection (LOD) for Pb, Hg, and Cd were 0.3 μg/dL, 0.1 μg/L, and 0.05 μg/L, respectively. The extraction and analysis procedures are described in detail in [32]. The quality control procedures for all analysis were in accordance with recommendations of the National Institute of Environmental Research [30].

### 2.3. Liver Function Biomarkers Measurement 

In order to estimate liver injury, we examined the serum AST, ALT, and GGT, which represented the degree of liver damage. For serum sample, whole blood is collected in a serum separation container separate serum tube (SST). Then, after conduction mixing, it is allowed to stand for 30 min, and the serum is separated by centrifugation. The total serum AST, ALT, and GGT were measured on an autoanalyzer ADVIA 1800 (Siemens Medical Solution, Malvern, PA, USA). The LOD values for liver biomarkers were 3.4, 4.2, and 4.0 U/L for AST, ALT, and GGT, respectively, and no value below the LOD was obtained for any participant. Procedures for extraction and analysis of liver function biomarkers are detailed in [33]. The cutoff values for the measured normal liver function biomarkers were calculated based on the reference range reported in [33], and these were as follows: serum AST < 34 U/L, serum ALT 10–49 U/L, serum GGT < 73 U/L (male), and serum GGT < 38 U/L (female).

### 2.4. Sociodemographic Characteristics

Sociodemographic characteristics included sex (male or female), age (19–29, 30–39, 40–49, 50–59, 60–69, and ≥70 years), and body mass index (BMI). BMI was calculated by dividing body weight in kilograms by the square of height and divided into categories of underweight (<18.5), normal (18.5–23), overweight (23–25), and obese (>25), according to the criteria of the Korean Society for the Study of Obesity (KOSSO). Smoking status was divided into three categories: non-smokers, former smokers, and current smokers. Three categories were used according to the state of alcohol consumption: non-drinkers, light drinkers (≤twice a month), and heavy drinkers (≥once a week to almost daily). Fish consumption frequency was also divided into three categories: non-consumption, sometimes (≥once a month to ≤3 times a week), and often (≥4 times a week to ≤3 times a day). HDL cholesterol levels associated with non-alcoholic fatty liver disease were divided into two categories: abnormal (HDL < 60) and normal (HDL ≥ 60).

### 2.5. Statistical Analysis

Considering that a stratified two-stage cluster sampling design was adopted in the present study, the selection probability was reflected, and the sample weights were applied for nonresponse. In addition, the stratification and cluster weights were included in the regression models. Since the concentration of heavy metals and three liver function biomarkers did not follow a normal distribution, logarithmic transformations were made. Values below the LOD of Pb, Cd, and Hg were replaced with values of the square root of the LOD. The concentrations of heavy metals were divided into quartiles for analysis. 

Multiple linear regressions were performed to evaluate the association between the heavy metals and liver function biomarkers of the individuals. Covariates were selected based on the association between the liver function biomarkers and demographic characteristics provided in previous epidemiological studies. The age, sex, BMI, smoking status, alcohol consumption, fish consumption frequency, HDL cholesterol levels associated with nonalcoholic fatty liver disease, and urinary creatinine were included as covariates in the regression models. In all analyses, a value of *p* < 0.05 was used for statistical significance, and the analyses were conducted using the Statistical Analysis System (SAS) 9.4 software.

### 2.6. Sensitivity Analysis

First, we performed a sensitivity analysis of 3699 participants to investigate the effect between heavy metal exposure and the whole range of liver function biomarkers. Second, in order to account for occupational exposure to heavy metals, we examined the association the between heavy metal exposure and liver function biomarkers after further adjusting for heavy metal-related occupations or not. This analysis was conducted with 67 participants who had occupational data.

## 3. Results

The demographic characteristics of the participants and the geometric mean (95% CI) of the heavy metal concentrations are presented in Table 1. The overall mean (±SE) age of the participants is 46.90 (±0.58). According to the data, Pb and Hg concentrations were significantly higher in men, while Cd concentrations were higher in women, and the heavy metal concentrations increased with age. The BPb and BHg concentrations show highest values of 1.88 (95% CI 1.80–1.96) and 3.05 (95% CI 2.86–3.25), respectively, in the 60–69- and 50–59-years age groups, while the UCd concentrations were highest at 0.59 (95% CI 0.51–0.68) in the ≤70 age group. The heavy metal concentrations increased with increasing BMI. The BPb and BHg concentrations were higher in smokers and drinkers, while UCd concentrations were higher in nondrinkers. With a higher frequency of fish consumption, there was a significant increase in BHg, UHg, and UCd concentrations. In addition, the proportion of participants with HDL cholesterol <60 (abnormal) is higher with increasing heavy metal concentrations. The concentration of liver function biomarkers according to the general characteristics of the study population is presented in the Supplementary Material (Supplemental Material, Appendix A). The geometric mean and concentration distributions of heavy metals and liver function biomarkers are presented in Table 2. The weighted geometric means (SE) of the serum AST, ALT, and GGT are 22.45 (0.15), 19.26 (0.17), and 18.02 (0.26) U/L, respectively. The weighted geometric means (SE) of the BPb and BHg are 1.57 (0.02) μg/dL and 2.65 (0.07) μg/L, respectively, while the corresponding GMs for UHg and UCd are 0.37 (0.01) and 0.42 (0.02) μg/L.

Proportional changes of liver function biomarkers by heavy metals after adjusting for potential confounders are presented in Table 3. For the BPb, BHg, and UCd, participants in the upper quartiles had higher serum AST, ALT, and GGT than those in the lowest quartile. There is evidence of a dose–response relationship between BPb and AST and GGT; BHg and AST; ALT and GGT; and UCd and ALT and GGT (*p* for trend <0.05). We could not observe any statistical differences between UHg and liver function biomarkers. 

Additionally, we examined the association between heavy metals and the whole range of liver function biomarker concentrations, and those results are similar except for UHg (Appendix A). After adjustment for heavy metal-related occupations, the results are consistent to those in non-adjustment models. (Appendix A).

## 4. Discussion

In the present study, data from the KoNEHS (2015–2017) were used to investigate the association between Pb, Cd, and Hg exposure and liver injury. The BPb, UCd, and BHg were associated with high serum AST, ALT, and GGT concentrations. However, the UHg showed no significant association with the liver function biomarkers. 

Regulatory efforts to reduce environmental exposure to heavy metals have been ongoing in Korea for several decades. Examples of activities such as the removal of Pb from gasoline, reducing paint Pb content, and smoking cessation programs have contributed to reducing such environmental exposure of heavy metals [34,35]. According the KoNEHS, despite the trend of decreasing heavy metal concentrations in Korean adults, they are higher than those in the United States and Canada, and higher or lower depending on the heavy metals for adults in Germany (Table 4) [29,31,36,37,38,39]. According to previous studies, factors such as age, sex, BMI, and lifestyle are related to liver function biomarkers as well as to heavy metal concentrations [8,40,41]. In Asian countries such as Korea, Japan, and China, it is reported that the high seafood consumption frequency contributes to heavy metals exposure [42,43,44,45]. In Korea, due to the geographic characteristics of islands or coastal countries, the country has characteristics of a food culture with high seafood consumption, which can affect exposure to heavy metals [31]. Several studies have reported that high concentrations of Pb, Cd, and Hg were detected in fish, shellfish, and crustacean consumption [46,47,48,49]. Considering that seafood consumption frequency is an important confounder of adults in Korea to heavy metals exposure, fish consumption frequency was considered a covariate in the present study. Heavy metal contamination of paddy soil is one of the serious concerns of Asian countries [50]. In particular, it has been reported that pollutants from abandoned mines in Korea contribute to heavy metal pollution in paddies through wastewater [51]. As a result, there is a risk of heavy metal exposure through the consumption of rice and grains that have accumulated heavy metals [52]. Therefore, the reason that the concentration of heavy metals in Asian countries including Korea is higher than in Western countries may be due to the high consumption of contaminated seafood and herbal medicines, as well as the consumption of grains such as rice [53,54]. Considering the BPb, the serum AST, ALT, and GGT concentrations for participants showing the highest BPb were significantly higher than those in the group with the lowest values. Our findings are consistent with previous epidemiological studies. Based on data from the National Health and Nutrition Examination Survey (NHANES), Obeng-Gyasi [24] demonstrated that serum AST, ALT, and GGT concentrations increase significantly with increasing BPb. Among the population in China, 158 in the group exposed to Pb and Cd showed 1.6 times higher serum GGT concentrations than the 109 controls [55]. In addition, in vivo studies reported increased serum AST, ALT, and GGT and liver injury in rats exposed to Pb [13,56]. In the UCd data, the serum AST, ALT, and GGT concentrations of the participants showing the highest UCd are significantly higher than those in the group with the lowest values. These results are in agreement with the findings of previous epidemiological studies. A study of Korean adults reported an increase in serum AST, ALT, and GGT concentrations with increased Cd exposure [14,16]. In an in vivo study, Almeida et al. [57] and Miandare et al. [15] suggested that increased serum AST and ALT concentrations due to Cd exposure cause hepatocytes death or damage. In addition, Kopeć et al. [58] and Varoni et al. [59] reported remarkably high serum ALT and AST concentrations in rats exposed to Cd. 

Concerning the BHg, the serum AST, ALT, and GGT concentrations for the participants showing the highest BHg are significantly higher than those in the group with the lowest values. In previous epidemiologic studies based on the Korea National Health and Nutrition Examination Survey (KNHANES) data, Moon [60] and Cave et al. [7] showed significant correlations between BHg and the serum ALT and GGT concentrations. In Lee et al. [61], the doubling of the BHg in a sample of the Korean population was related to the increase of AST and ALT concentrations by 0.676 and 1.067 times, respectively. In vivo studies in rats reported an increase in serum AST, ALT, and GGT concentrations and liver tissue necrosis in rats exposed to Hg [62,63]. These results are consistent with our findings. Although the UHg showed no significant association with liver function markers, according to several studies, UHg may not be an appropriate indicator to assess the role of liver function. This is because Hg is preferentially excreted in the feces through the intestinal system [64]. Comprehensively, the results of the present study demonstrate the relationships between heavy metals exposure and elevated liver function biomarkers. 

Although the hepatotoxicity mechanisms of Pb, Cd, and Hg remain unclear, their ability to produce reactive oxygen species (ROS) responsible for oxidative stress and the destruction of the antioxidant defense system is an important concept [65,66,67,68]. Heavy metals deplete cells of thiol-containing antioxidants and enzymes, and induce increased production of ROS, such as hydroxyl radicals (HO), hydrogen peroxide (H_2_O_2_), or superoxide radicals (O_2_^−^) [65]. Several studies showed that heavy metals exposure promotes ROS production, and ROS promotes the production of peroxides, which contribute to cell membrane damage [9,65,69,70]. It has been reported that this causes an imbalance between the synthesis and degradation of enzymatic proteins, thereby releasing liver enzymes in the blood stream because of hepatic necrosis [62]. In addition, increased ROS production can overwhelm the cells intrinsic to antioxidant defenses, resulting in oxidative stress [65]. Oxidative stress, which occurs when the balance between the antioxidant defense system and the free radical generating system is disrupted, can contribute to several diseases [71]. Consequently, it is suggested that metal-induced oxidative stress may have toxic effects on cells [65,72].

It has been reported that Pb binds to the –SH group of glutathione (GSH), thereby reducing the antioxidant activity and increasing the ROS production [65,73,74,75]. Sandhir and Gill [76] and Omobowale et al. [77] observed that the depletion of GSH and the activity of the antioxidant defense system are significantly reduced in rats exposed to Pb. In addition, several studies indicated that exposure to Cd enhances ROS production by binding to the sulfhydryl groups of the protein and depleting glutathione [78,79,80,81]. Bagchi et al. [82] and Dwivedi et al. [83] reported the depletion of GSH; lipid peroxidation; other cellular damage, including DNA damage; and increased ROS in Cd-exposed rats as evidence of oxidative stress. Hg has been reported to induce oxidative stress and ROS production by binding to the –SH group of glutathione and protein [84,85,86]. In vivo studies have shown that the accumulation of oxidative stress is due to mercury exposure and the destruction of the balance between ROS and the antioxidant defense system [87,88].

The limitations of the present study are as follows. First, because the KoNEHS is a cross-sectional study, an accurate temporal causal association between heavy metals exposure and changes in liver biomarkers could not be established. Therefore, the biological implications of the statistical association should be interpreted with caution. Second, blood heavy metals, which reflect relatively recent exposure compared to urine heavy metals, have limitations in explaining the burden on the body due to long-term exposure. Therefore, caution is required interpreting the results. Third, although patients with chronic hepatitis, acute liver disease, fatty liver disease, and liver cancer were excluded during screening, participants with mild hepatic dysfunction associated with other causative factors were probably included in the study. Fourth, although this study focused on exposure to Pb, Cd, and Hg, there may be concomitant exposure to other metals that can induce oxidative stress related to a decrease of liver function. Fifth, because of the limitations of the KoNEHS survey components, an integrated liver function assessment was impossible. However, this study has several distinct strengths. First, the results of this study can be interpreted as the result of the national population because data from KoNEHS, which represents the Korean adult population, were used. This study was the first to evaluate the relationship between Pb, Hg, and Cd exposure and liver injury in the Korean population. The results of our study provide reliable epidemiological evidence regarding the liver function effects of heavy metals at the current level of exposure.

## 5. Conclusions

Our results are unique in that they are based on a nationally representative population with the largest sample size. Our observations clearly showed that Pb, Cd, and Hg at the current levels were significantly association with serum AST, ALT, and GGT. These results provide evidence that heavy metals exposure may result in liver injury. To elucidate the potential effect of heavy metals exposure, further studies should be performed.

## Figures and Tables

**Table 1 ijerph-18-06783-t001:** The descriptive statistics of demographic characteristics and geometric means (95% CI) of heavy metal concentrations in participants.

	*n* (%)		BPb (μg/dL)		BHg ^a^ (μg/L)		UHg ^a^ (μg/L)		UCd (μg/L)	
**Total**	2953	(100)	1.57	(1.53–1.62)	2.65	(2.53–2.79)	0.37	(0.35–0.39)	0.42	(0.39–0.45)
**Sex**										
Male	1205	(40.81)	1.82	(1.75–1.89)	3.09	(2.58–3.34.)	0.4	(0.38–0.44)	0.40	(0.37–0.44)
Female	1748	(59.19)	1.38	(1.34–1.43)	2.33	(2.23–2.43)	0.35	(0.33–0.37)	0.43	(0.40–0.47)
*p-*value			<0.001		<0.001		<0.001		0.288	
**Age group (years)**										
19–29	231	(7.82)	1.27	(1.17–1.37)	1.92	(1.73–2.13)	0.35	(0.30–0.40)	0.25	(0.22–0.29)
30–39	411	(13.92)	1.45	(1.38–1.52)	2.78	(2.58–2.99)	0.41	(0.37–0.45)	0.34	(0.31–0.38)
40–49	500	(16.93)	1.53	(1.47–1.59)	2.96	(2.76–3.17)	0.40	(0.37–0.44)	0.41	(0.37–0.46)
50–59	680	(23.03)	1.83	(1.74–1.97)	3.05	(2.86–3.25)	0.38	(0.35–0.41)	0.53	(0.48–0.59)
60–69	712	(24.11)	1.88	(1.80–1.96)	3.03	(2.79–3.30)	0.36	(0.33–0.39)	0.59	(0.53–0.65)
≤70	419	(14.19)	1.72	(1.61–1.83)	2.37	(2.13–2.65)	0.33	(0.30–0.36)	0.59	(0.51–0.68)
*p*-value			<0.001		<0.001		<0.001		<0.001	
**BMI (kg/m^2^)**										
<18.5	74	(2.51)	1.32	(1.18–1.48)	1.78	(1.47–2.15)	0.31	(0.25–0.38)	0.28	(0.22–0.36)
18.5–23	996	(33.73)	1.43	(1.37–1.50)	2.35	(2.20–2.50)	0.35	(0.33–0.38)	0.37	(0.34–0.41)
23–25	771	(26.11)	1.61	(1.54 -1.68)	2.89	(2.69–3.10)	0.37	(0.34–0.40)	0.45	(0.41–0.50)
>25	1112	(37.66)	1.74	(1.68–1.82)	2.96	(2.79–3.14)	0.41	(0.38–0.43)	0.46	(0.43–0.50)
*p*-value			<0.001		<0.001		<0.001		<0.001	
**Smoking status**										
Never	1974	(66.85)	1.42	(1.38–1.45)	2.39	(2.28–2.51)	0.35	(0.33–0.37)	0.40	(0.37–0.44)
Former	553	(18.73)	1.87	(1.78–1.96)	3.37	(3.11–3.65)	0.43	(0.39–0.48)	0.47	(0.42–0.52)
Current	426	(14.43)	1.95	(1.84–2.01)	3.09	(2.76–3.46)	0.42	(0.38–0.46)	0.43	(0.39–0.48)
*p*-value			<0.001		<0.001		<0.001		0.005	
**Drinking status**										
Never	1003	(33.97)	1.50	(1.44–1.56)	2.37	(2.25–2.50)	0.33	(0.31–0.35)	0.46	(0.41–0.51)
Light	1020	(34.54)	1.45	(1.40–1.51)	2.45	(2.29–2.63)	0.35	(0.33–0.38)	0.44	(0.41–0.48)
Heavy	930	(31.49)	1.77	(1.70–1.84)	3.14	(2.93–3.37)	0.44	(0.40–0.47)	0.37	(0.34–0.41)
*p*-value			<0.001		<0.001		<0.001		<0.001	
**Fish consumption frequency**										
Rarely	287	(9.72)	1.55	(1.39–1.72)	1.80	(1.57–2.08)	0.33	(0.29–0.37)	0.40	(0.41–0.51)
Sometimes	2469	(83.61)	1.57	(1.53–1.62)	2.75	(2.63–2.88)	0.38	(0.36–0.41)	0.42	(0.41–0.48)
Often	197	(6.67)	1.67	(1.52–1.83)	3.44	(3.00–3.94)	0.32	(0.28–0.37)	0.49	(0.34–0.40)
*p*-value			0.178		<0.001		0.001		0.040	
**HDL-Cholesterol (mg/dL)**										
HDL < 60	1925	(65.19)	1.64	(1.58–1.80)	2.70	(2.55–2.86)	0.38	(0.36–0.40)	0.45	(0.42–0.49)
HDL ≥ 60	1028	(34.81)	1.47	(1.41–1.54)	2.59	(2.43–2.76)	0.36	(0.34–0.39)	0.37	(0.34–0.40)
*p*-value			<0.001		0.089		0.131		<0.001	

^a^ Total mercury. Abbreviations: CI, confidence interval; BPb, blood lead; BHg, blood mercury; UHg, urinary mercury; UCd, urinary cadmium; BMI, body mass index; HDL, high-density lipoproteins cholesterol.

**Table 2 ijerph-18-06783-t002:** Distributions of heavy metals and liver function biomarker concentrations among participants (*n* = 2593).

	GM	SE of GM	95% CL for GM	Percentile				
				25th	50th	75th	99th	Max
**Liver function biomarkers**								
AST (U/L)	22.45	0.15	(22.16–22.74)	19.08	22.06	25.56	32.71	34.00
ALT (U/L)	19.26	0.17	(18.93–19.60)	14.59	18.30	23.42	44.50	49.00
GGT (U/L)	18.02	0.26	(17.51–18.55)	11.79	16.47	25.24	64.67	73.00
**Heavy metals**								
BPb (μg/dL)	1.57	0.02	(1.53–1.62)	1.17	1.56	2.13	5.09	14.82
BHg (μg/L) ^a^	2.65	0.07	(2.53–2.79)	1.71	2.64	4.03	13.94	125.49
UHg (μg/L) ^a^	0.37	0.01	(0.35–0.39)	0.22	0.32	0.58	3.05	8.70
UCd (μg/L)	0.42	0.02	(0.39–0.45)	0.22	0.42	0.80	2.92	16.81

^a^ Total mercury. Abbreviations: CL, confidence level; SE, standard error; GM, geometric mean; BPb, blood lead; BHg, blood mercury; UHg, urinary mercury; UCd, urinary cadmium; AST, alanine aminotransferase; ALT, aspartate aminotrans ferase; GGT, gamma-glutamyl transferase.

**Table 3 ijerph-18-06783-t003:** Adjusted proportional changes (95% CI) between heavy metals and liver function biomarkers (*n* = 2593).

			AST ^a^		ALT ^a^		GGT ^a^	
		*n*	Exp β	(95% CI)	Exp β	(95% CI)	Exp β	(95% CI)
**BPb (μg/dL)**								
Q1	(0.33–1.26)	737	1.000	(Reference)	1.000	(Reference)	1.000	(Reference)
Q2	(1.26–1.71)	739	1.021	(0.994–1.049)	1.047	(1.007–1.089)	1.015	(0.961–1.073)
Q3	(1.71–2.31)	738	1.024	(0.998–1.051)	1.031	(0.986–1.077)	1.070	(1.007–1.137)
Q4	(2.31–14.82)	739	1.042	(1.012–1.072)	1.047	(1.001–1.096)	1.098	(1.032–1.169)
*p* for trend			0.005		0.101		<0.001	
**BHg (μg/L) ^b^**								
Q1	(0.33–1.86)	738	1.000	(Reference)	1.000	(Reference)	1.000	(Reference)
Q2	(1.86–2.81)	735	1.028	(1.004–1.052)	1.054	(1.012–1.097)	1.011	(0.962–1.063)
Q3	(2.81–4.42)	742	1.010	(0.985–1.036)	1.051	(1.012–1.092)	1.075	(1.019–1.134)
Q4	(4.43–125.49)	738	1.044	(1.016–1.073)	1.119	(1.074–1.165)	1.184	(1.112–1.260)
*p* for trend			0.009		<0.001		<0.001	
**UHg (μg/L) ^b^**								
Q1	(0.10–0.22)	725	1.000	(Reference)	1.000	(Reference)	1.000	(Reference)
Q2	(0.23–0.33)	764	1.014	(0.992–1.036)	1.039	(0.998–1.083)	1.047	(0.996–1.101)
Q3	(0.34–0.61)	732	0.996	(0.969–1.024)	1.014	(0.969–1.062)	1.020	(0.962–1.081)
Q4	(0.62–8.70)	732	1.003	(0.976–1.031)	1.022	(0.975–1.072)	1.047	(0.991–1.105)
*p* for trend			0.934		0.586		0.123	
**Ucd (μg/L)**								
Q1	(0.05–0.23)	739	1.000	(Reference)	1.000	(Reference)	1.000	(Reference)
Q2	(0.23–0.47)	736	1.024	(0.997–1.051)	1.045	(1.001–1.091)	1.014	(0.952–1.079)
Q3	(0.47–0.88)	741	1.008	(0.978–1.039)	1.076	(1.022–1.132)	1.049	(0.989–1.113)
Q4	(0.88–16.81)	737	1.034	(1.001–1.069)	1.092	(1.031–1.156)	1.085	(1.017–1.157)
*p* for trend			0.125		0.001		0.022	

^a^ AST < 34 U/L, serum ALT 10–49 U/L, serum GGT < 73 U/L (male), and serum GGT < 38 U/L (female); ^b^ Total mercury adjusted for sex, age, smoking status, drinking status, BMI, fish consumption frequency, and HDL. Abbreviations: BPb, blood lead; BHg, blood mercury; UHg, urinary mercury; UCd, urinary cadmium; AST, alanine aminotransferase; ALT, aspartate aminotrans ferase; GGT, gamma glutamyl transferase.

**Table 4 ijerph-18-06783-t004:** Comparison of heavy metal concentrations among the general population.

	Geometric Mean					
	KoNESH 3	KoNEHS 2 ^a^	KoNEHS 1 ^b^	NHANES ^c^	CHMS ^d^	GerES ^e^
BPb (μg /dL)	1.60	1.94	1.79	1.23	1.37	3.07
BHg (μg/L) ^f^	2.75	3.11	3.08	0.71	0.76	0.58
UHg (μg/L) ^f^	0.35	0.38	0.53	0.49	0.22	0.43
UCd (μg/L)	0.36	0.38	0.58	0.22	0.35	0.23

^a^ KoNEHS, aged 19 years and over (Choi et al., 2017); ^b^ KoNEHS, aged 19 years and over (Park et al., 2016); ^c^ NHANES, aged 20 years and over (Park et al., 2013; Xu et al., 2021); ^d^ CHMS, aged 6 years and over (Health Canada, 2010); ^e^ GerES, aged 18 years and over (Becker et al., 2002); ^f^ Total mercury.

## Data Availability

This study used data from the Second Korean National Environmental Health Survey (KoNEHS) which was conducted by Ministry of Environment, National Institute of Environmental Research. The data presented in this study are available on request from the corresponding author. The data are not publicly available due to protect personal information.

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
