# Peer review of "Association between Pb, Cd, and Hg Exposure and Liver Injury among Korean Adults"

_ijerph, 2021, doi:10.3390/ijerph18136783_

Round 1

Reviewer 1 Report

General comment:

The manuscript brings new and useful information about the association between three heavy metals (Pb, Cd and Hg) exposure and liver function among Korean adults. Authors investigated the associations between levels of blood or urine heavy metals and liver function biomarkers such as aspartate aminotransferase (AST), alanine aminotransferase (ALT), and gamma-glutamyl transferase (GGT) using multiple linear regression analysis. The results are interesting, and the authors concluded that Pb, Cd and Hg exposures decreased liver function. This study findings potentially could contribute to the research field. However, some minor concerns require to be addressed to clarity and improve the present version of the manuscript.

Specific comments:

Title

The title is informative and relevant to the major findings. But, there are some uppercase letters in the title.

Abstract

In the abstract, the aim of the study clearly mentioned and major results also presented. 

Introduction

The research objective is clearly outlined. In line-35, the 2nd sentence of the introduction section is not appropriate, authors should rewrite the sentence.  

Materials and Method

Overall well described. But the “2.2. Heavy metals measurement” section lacks a proper description of QA and QC.  In the section “2.3. Liver function biomarkers measurement” authors should add proper references.

 Results

The results are not well described. In line-166, “The BPb and BHg concentrations were higher in smokers and drinkers, while UCd concentrations were higher in non-drinkers”. Why? please explain a little about it. The description of results from table 3 is not interesting and monotonous. The authors should improve the description of results from table 3.

Discussion

Overall good and organized.

Conclusion

Conclusion properly answered the aims of the study and was supported by the results. Major limitations and opportunities to inform future research are addressed.  

Reviewer 2 Report

Abstract and beyond: I would use the term liver injury rather than function. Since these markers are an indication of injury.

Introduction:

Line 35 and beyond… is found in gasoline etc… not, is released the…

In this study why did you not examine Alkaline Phosphatase and total bilirubin, Albumin  etc?

Methods:

Well done.

 Discussion:

Speak more to metals exposure in Korea.. . What are the sources that would bring about these physiological changes?

Limitations, speak to the half-life of these metals in blood, urine etc. and how that limits interpretation  of exposure risk.

Author Response

This manuscript is a resubmission of an earlier submission. The following is a list of the peer review reports and author responses from that submission.